# Molecular Investigation of *Klebsiella pneumoniae* from Clinical Companion Animals in Beijing, China, 2017–2019

**DOI:** 10.3390/pathogens10030271

**Published:** 2021-02-27

**Authors:** Zhenbiao Zhang, Lei Lei, Haixia Zhang, Hegen Dai, Yu Song, Lei Li, Yang Wang, Zhaofei Xia

**Affiliations:** 1College of Veterinary Medicine, China Agricultural University, Beijing 100193, China; zbzhang2017@cau.edu.cn (Z.Z.); axiastar@cau.edu.cn (H.Z.); B20183050429@cau.edu.cn (H.D.); ps20183050834@cau.edu.cn (Y.S.); jihyo@cau.edu.cn (L.L.); 2Beijing Key Laboratory of Detection Technology for Animal-Derived Food Safety, China Agricultural University, Beijing 100193, China; leilei910@zafu.edu.cn; 3College of Animal Science and Technology, College of Veterinary Medicine, Zhejiang A & F University, Hangzhou 311300, China

**Keywords:** *Klebsiella pneumonia*, companion animals, carbapenem-resistant, hypervirulent, multidrug-resistant

## Abstract

This work is aimed to elucidate the prevalence and characteristics of antimicrobial resistance, virulence, and molecular typing in *Klebsiella pneumoniae* from clinical companion animals in Beijing, China. In total, 105 *K. pneumoniae* (2.0%) isolates were recovered from 5359 samples (dogs, *n* = 3356; cats, *n* = 2003). All tested isolates exhibited high resistance to amoxicillin-clavulanate (74.3%). Moreover, resistance rates in dog isolates (2.1%) were significantly higher than in cat isolates (0.9%); however, the rate of multidrug-resistance (MDR) was 57.1% and the MDR prevalence in cats was significantly higher than dogs. Whole-genome sequencing demonstrated plasmids IncX4 and IncFIA (HI1)/FII(K) carried *mcr-1* (*n* = 1) and *mcr-8* (*n* = 1), but *bla*_OXA-181_ (*n* = 1) and *bla*_NDM-5_ (*n* = 4) were harbored in IncX3-type plasmids, and the above genes were in different isolates. The most prevalent sequence types (STs) in companion animals were ST1 (*n* = 9) and ST37 (*n* = 9). Compared to National Center for Biotechnology Information (NCBI) data on human *K. pneumoniae*, resistance genes *bla*_CTX-M_ and *bla*_TEM_ were more prevalent in human isolates; however, *aac(6′)-Ib-cr* and *oqxAB* showed a higher prevalence in companion animals. Hypermucoviscosity was reported in 9 (8.6%) isolates, whereas 64 isolates (61.0%) were hypervirulent *K. pneumoniae* (hvKP) via the *Galleria mellonella*. These findings validate the high risk of *K. pneumonia* and necessitate its relevant control in pet clinics.

## 1. Introduction

*Klebsiella pneumoniae* is an opportunistic pathogen that colonizes the skin, upper respiratory tract, and digestive tract of healthy asymptomatic subjects [1]. Also, it is a primary cause of diseases in neonates, elderly and immunocompromised humans and animals, including pneumonia, wound infections, urinary tract infections (UTIs), sepsis and meningitis [2]. *K. pneumoniae* is inherently resistant to penicillin; members of this population, in most cases, acquire resistance to multiple antibiotics. Thus, it is associated with resistance to important antimicrobial agents, thereby a significant threat to public health [3].

In recent years, *K. pneumoniae* has rapidly become a multidrug-resistant (MDR) pathogen. It develops resistance to third-generation cephalosporins, fluoroquinolones, and aminoglycosides. Of concern, *K. pneumoniae* has increasingly become resistant to carbapenems by acquiring carbapenemases [4]. The World Health Organization recognizes extended-spectrum β-lactam (ESBL)-producing and carbapenem-resistant *K. pneumoniae* (CRKP) as a serious public health threat [5]. Colistin and tigecycline were considered the “last resort” in managing CRKP-related critical carcinogens. However, with the increase in CRKP prevalence, various carbapenem antibiotics have proven ineffective [6]. Companion animals had been reported to harbor *bla*_NDM_, which could quickly contaminate wild birds through dogs, flies, and livestock as hosts; this phenomenon is a human health threat [7]. The first mobile colistin resistance gene, *mcr-1*, was found in Enterobacteriaceae (mainly *Escherichia coli* and *Klebsiella pneumoniae*) [8], other variants of *mcr* were also discovered, and have spread widely in different hosts and regions [9,10].

Based on previous reports, hypervirulent *K. pneumoniae* (hvKP) can cause acquired liver abscess with severe organ failure. Notably, 22.8% of *K. pneumoniae* clinical isolates were identified as hvKP in China [11]. Various hypervirulence-associated factors are critical in hvKP isolates, including capsular serotypes (K1 and K2), sequence types (ST23), a virulence plasmid, a pathogenicity island, and several virulence factors [12]. The hypermucoviscosity *K. pneumoniae* (hmKP) is attributed to the elevated production of capsular polysaccharides encoded by specific virulence genes, *rmpA* and *rmpA2* [13]. Otherwise, a genetic and phenotypic convergent clone, CR-hvKP, simultaneously exhibits carbapenem resistance and hypervirulence has emerged in recent years [14].

*K. pneumoniae* isolates had been shown to elevate the risk of antibiotic treatment failure both in humans and companion animals [2]. Similarly, if the isolates are transmitted to humans through pets, the antimicrobial bacteria present in companion animals may significantly impact human public health. The *K. pneumoniae* isolate that infects companion animals potentially belongs to high-risk clonal lineages found in humans [15]. As a result, there is an urgent need to understand the molecular and genetic characteristics of *K. pneumoniae* isolates from a veterinary medicine and public health perspective. However, the current prevalence and characteristics of *K. pneumoniae* in clinical companion animals in China is unknown. In this study, we clarified the prevalence and characteristics of antimicrobial resistance, virulence, and molecular typing in *K. pneumoniae* from clinical companion animals in Beijing, China.

## 2. Results

### 2.1. Pets Samples and K. pneumoniae Isolates

We collected 5359 samples from 3356 dogs and 2003 cats, and isolated 105 *K. pneumoniae* isolates (2.0%, 95% confidence interval (CI): 1.6–2.4, *n* = 105/5359); 85 from dogs (2.5%, 95% CI: 2.0–3.1, *n* = 85/3356) and 20 from cats (1.0%, 95% CI: 0.6–1.5, *n* = 20/2003). A significant difference was noted in the separation rate between dogs and cats (*p* < 0.05). Throat swabs (6.5%, 95% CI: 3.5–10.9, *n* = 13/200), nasal swabs (4.6%, 95% CI: 2.1–8.6, *n* = 9/194), and tracheal lavage (6.3%, 95% CI: 2.8–12.1, *n* = 8/126) exhibited higher rates of bacterial detection than the total rates (*p* < 0.05) (Table 1).

### 2.2. Antimicrobial Resistance

*K. pneumoniae* isolates exhibited the highest (74.3%, *n* = 78/105) resistance to amoxicillin-clavulanate, high resistance to doxycycline (52.4%, *n* = 55/105) and trimethoprim-sulfamethoxazole (57.1%, *n* = 60/105). However, *K. pneumoniae* isolates demonstrated relatively low resistance to colistin (7.6%, *n* = 8/105), ceftazidime-avibactam (4.8%, *n* = 5/105), imipenem (4.8%, *n* = 5/105), and meropenem (3.8%, *n* = 4/105) (Appendix A). Of these isolates, resistance of isolates in dogs (2.1%, 95% CI: 1.6–2.6, *n* = 70/3356) was significantly higher than that of cat isolates (0.9%, 95% CI: 0.6–1.5, *n* = 19/2003) (*p* < 0.05). Notably, the resistance of cat isolates to five antimicrobial agents tested was significantly higher than that of dogs (*p* < 0.05) (Figure 1A, Appendix A). Minimum inhibitory concentrations MIC_50_ and MIC_90_ of cats were generally above or equal to that of dogs. Overall, 60 isolates showed MDR (57.1%, *n* = 60/105). The MDR prevalence rate of cats was significantly higher than dogs (*p* < 0.05). Compared to the MDR rate between February and August 2018, that between April to October 2019 was significantly lower (*p* < 0.05) (Figure 1B).

### 2.3. Antibiotic Resistance Genes and Virulence Genes

The mobile colistin resistance gene *mcr-1* (Kp141) and *mcr-8* (Kp24) were detected in *K. pneumoniae* from tracheal lavage and urine from different cats. Carbapenem resistance genes *bla*_OXA-181_ and *bla*_NDM-5_ were harbored in one (Kp3), and four (Kp79, Kp84, Kp165, Kp181) isolates, respectively. Additionally, we analyzed the sequence read archive (SRA) sequences (*n* = 46) of human *K. pneumoniae* (HKp) from National Center for Biotechnology Information (NCBI) to demonstrate the similarities, differences, and relevance of the molecular characteristics of *K. pneumoniae* from humans and companion animals. Resistance genes, *mcr* and *bla*_NDM_, were not present in the genomes of *K. pneumoniae* isolated from humans; other variants were also absent. *bla*_KPC_ (95.7%, *n* = 44/46) was harbored in human isolates, whereas none was found in companion animals. Resistance genes *bla*_CTX-M_ and *bla*_TEM_ were more prevalent among *K. pneumoniae* isolates from humans, compared to companion animals (*p* < 0.01). Meanwhile, isolates from companion animals showed higher rates of *aac(6′)Ib-cr* and *oqxAB* than those from humans (*p* < 0.01). Moreover, the prevalence of *bla*_CTX-M_ and *rmtB* from cats was higher than in dogs (*p* < 0.05); notably, *bla*_SHV_ (87.6%, *n* = 92/105) was the most prevalent β-lactamase resistance gene in companion animals (Figure 2). Besides, we found eight CTX-M-genotypes (-3, -14, -15, -27, -55, -64, -65, -104) in companion animals, dominated by *bla*_CTX-M-55_ (14.3%, *n* = 15/105). *bla*_CTX-M-65_ (60.9%, *n* = 28/46) was the largest proportion of four CTX-M-genotypes (-14, -15, -65, -147) in human isolates. Among them, companion animals harbored *bla*_CTX-M-55_ (*p* < 0.05) and *bla*_CTX-M-65_ (*p* < 0.01), which were significantly more and less prevalent than humans, respectively. The fosfomycin resistance gene *fosA* showed full coverage in companion animals and human isolates (Figure 2).

Similarly, virulence-associated genes enterotoxins (*entA/B/E/S*), ferrienterochelin receptor (*fepA/B/C*), fimbriae (*fimA/E*), outer membrane protein (*ompA*), and common pili (*ecpA/B/C/D/E/R*) were harbored by all isolates from humans and companion animals, as revealed using abricate (Figure 2). The *ybtA/E/P/Q/T/U/X*, *iucA/B*, *rmpA2*, *fepD*, *fyuA*, *irp1/2*, and *iutA* from human isolates were significantly more prevalent (*p* < 0.05) than those from companion animals; however, virulence genes of dogs and cats exhibited no difference. Interestingly, the ST23-hvKP (*n* = 5) isolates harbored *ybt*, *clb*, *iro*, *iuc*, *ent*, *rmpA*, and *rmpA2* virulence genes, with the least resistance genes. Moreover, ST37-hvKP isolates harbored common virulence factors *ent*, *fep*, *fim*, *ompA*, and *ecp* but simultaneously took along more resistance genes (Figure 2).

### 2.4. Hypervirulent, Hypermucoviscosity, Capsule Serotype, and O-Antigen of K. Pneumonia

Of the 105 isolates, 64 (61.0%, *n* = 64/105) demonstrated higher or equal pathogenicity to the positive control ATCC43816 after injection for 72 h in *G. mellonella*, demonstrating that these isolates were hypervirulent (Appendix A). ST37 (12.5%, *n* = 8/64), ST1 (7.8%, *n* = 5/64) and ST23 (7.8%, *n* = 5/64) were predominately among hvKP. There were 50 (58.8%, *n* = 50/85) and 14 (70%, *n* = 14/20) hvKP isolates from dogs and cats, respectively, but showed no significant difference. Mucoid was observed in nine (8.6%, *n* = 9/105) isolates, evaluated by mucoviscosity assay (Figure 3); however, three of them were not among hvKP. There was no association between hvKP and hmKP (*p* > 0.05). The resistance and virulence of mucoid and non-mucoid isolates had no significant difference in this study (*p* > 0.05). In addition, there was no significant difference in antibiotic resistance between hypervirulence and hypovirulence isolates from companion animals (*p* > 0.05). MDR-hvKP accounted for 39.0% (*n* = 41/105) of total *K. pneumonia* isolates, harboring four CR-hvKP. Additionally, capsule serotype and O-antigen type were obtained from Kleborate. Among the six hv-KP harboring *ybt*, *clb*, *iro*, *iuc* and *rmpA*, sequence types and capsular serotypes of Kp16, Kp21, Kp139, Kp175, and Kp177 were ST23 and KL1, but Kp12 was ST65 and KL2, respectively (Figure 3). The KL1 and KL2 isolates were hmKP, and the O-antigen type of 77.8% (*n* = 7/9) hmKP was O1v2. In companion animals, we obtained 47 capsule serotypes and 10 O-antigen types; the most prevalent serotypes were KL19 (10.5%, *n* = 11/105) and O1v2 (20.0%, *n* = 21/105). In human isolates, the most abundant serotypes were KL47 (65.2%, *n* = 30/46) and O2v1 (43.5%, *n* = 20/46), respectively (Figure 3). O loci named OL101 onwards were defined based on gene content and are not yet associated with a specific serologically defined O type.

### 2.5. Diversity Genotypes of the K. pneumoniae Isolates

Most of the 105 isolates belonged to the *KpI* phylogroup (87.6%, *n* = 92/105), with only a few isolates belonging to *KpIIa* (1.9%, *n* = 2/105), *KpIII* (6.7%, *n* = 7/105) and *KpIIb* (3.8%, *n* = 4/105) in companion animals. MLST presented a diverse distribution, 89 out of 105 *K. pneumoniae* isolates were assigned to 48 known STs, whereas 16 isolates represented 16 (25.0%, *n* = 16/64) novel STs (ST4565-ST4567, ST4570, ST4572-ST4580, ST4874-ST4876) (Figure 2 and Figure 4). However, STs of all human isolates were classified as ST11, indicating that ST11 is of great concern and widely reported in humans. We also reported two ST11 animal isolates shared by the dog (Kp138) and cat (Kp164), they were not in the same evolutionary branch as human isolates in the core SNP-based phylogenetic tree (Figure 2). In companion animals, the most prevalent STs were ST1 (8.6%, *n* = 9/105) and ST37 (8.6%, *n* = 9/105), followed by ST15 (*n* = 6) and ST23 (*n* = 5). ST1 and ST37 were prevalent in cats (*n* = 3) and dogs (*n* = 7), respectively. The minimum spanning trees of MLST further validated the commonness of *K. pneumoniae* isolates from humans and companion animals with the same STs (Figure 4). In this study, the STs of four *bla*_NDM-5_-positive *K. pneumoniae* were ST1 (Kp79, Kp84, Kp165) and ST307 (Kp181), but the isolates harboring *mcr-1* (Kp141), *mcr-8* (Kp24) and *bla*_OXA181_ (Kp3) were ST656, ST3410, and ST16.

### 2.6. Characterization of Plasmids

After assembling backbone sequences, all contigs and gaps were identified by whole-genome analysis. Among the Inc-type plasmids, IncFIB (67.6%, *n* = 71/105) was prevalent in *K. pneumoniae* isolates of companion animals, whereas human isolates were covered by IncFII (100%, *n* = 46/46). Of these, IncFIA was significantly more prevalent in companion animals than that of humans (*p* < 0.01), but the prevalence of IncFII and IncR were significantly higher in human isolates (*p* < 0.01). All Inc-type plasmids exhibited no difference between dogs and cats (Figure 2). Plasmids IncX3, IncX4, and IncFIA (HI1)/FII(K) harbored *bla*_OXA-181_ (Kp3), *mcr-1* (Kp141) and *mcr-8* (Kp24) respectively. Moreover, BLASTn results demonstrated that *bla*_NDM-5_ was harbored in four isolates of companion animals (Figure 2 and Figure 5). The *bla*_NDM-5_ gene was distributed among dogs (*n* = 2) and cats (*n* = 2), source from urine (*n* = 3) and abscess (*n* = 1). The complete genome sequences of four *bla*_NDM-5_-positive isolates contained regions showing > 99% nucleotide sequence homology to the reference plasmid pNDM_MGR194 (46253bp, accession No. KF220657), suggesting that *bla*_NDM-5_ were likely located on IncX3-type plasmids (Figure 5A). Also, the *bla*_NDM-5_ gene was included in an insertion sequence (IS) cassette (∆IS*Aba125*-IS*5*-*bla*_NDM_-*ble*-*trpF*-*dsbC*-IS*26*) compared using ISfinder. Similarly, the plasmids of IncX3 and IncX4 which harbored *bla*_OXA-181_ and *mcr-1* were consistent with reference plasmids pABC239-OXA-181 (51479bp, accession No. MK412916) and pECGD-8-33 (33307bp, accession No. KX254343), respectively (Figure 5B,C). Interestingly, the *mcr-8* gene was harbored by IncFIA (HI1)/FII(K), similar to plasmid p18-29mcr-8.2 (91072bp, accession No. MK262711) within the structure of IS*Ecl1*-*mcr-8.2*-*orf*-IS*Kpn26* (Figure 5D).

## 3. Discussion

*K. pneumoniae* is an important host and transmission carrier of clinically important antimicrobial resistance genes in humans and animals [16]. Typically, humans have close contact with pets; therefore, there is a close association between the health of both. However, there is currently a dearth of clinical research on *K. pneumoniae* from companion animals in China. Herein, we report the prevalence of antibiotic resistance, virulence, molecular typing, and phylogroups in *K. pneumoniae* from companion animals in Beijing, China. Previous findings demonstrated that the overall prevalence of recovered (2.0%) *K. pneumoniae* was slightly lower than 3.53% in Italy [17]. *K. pneumoniae* is the primary pathogen of UTIs, and is usually associated with resistance to the most significant antibiotics [15]. Urine (53.7%) accounted for the largest proportion (1.3%) of samples, from which the largest number of *K. pneumoniae* isolates (*n* = 37) was isolated in this study, this was similar with a previous study in Japan [18].

Among the *K. pneumoniae* isolates from companion animals, the resistance rate in dogs was significantly higher than in cat isolates, but showed no difference in South Korea [19]. This discordance in results could be explained by different types and quantities of tested drugs. An overall MDR rate of 57.1% was slightly higher than that previously reported in Singapore (50.0%) [20]. Otherwise, the MDR prevalence in cats was significantly higher than in dogs in this study (*p* < 0.05). As a result, this aggravates the threat to human health, as owners are more likely to exhibit intimate behavior with companion animals, increasing the probability of mutual transmission. Colistin had been used as an animal feed contributing to the prevalence of *mcr* in China; similarly, studies have reported wide prevalence in Vietnam and other South Asian countries. However, the prevalence of *mcr-1* in food animals in Europe and America is low [21], which may be attributed to the fact that countries in these regions have not approved colistin use as an antibacterial growth-promoting agent. Elsewhere, a previous study demonstrated low prevalence (<1%) of *mcr-1* of *K pneumoniae* from humans [22]. Moreover, *mcr-1* in *K. pneumoniae* from companion animals; thus, the pet food industry was speculated to be a source of *mcr-1* [23]. Since the first description of *mcr-8,* with IncFII-type plasmid as the carrier, it has been widely disseminated among *K. pneumoniae* isolates of livestock origin [6]. In the current study, *mcr-1* (*n* = 1) and *mcr-8* (*n* = 1) were carried by plasmids IncX4 and IncFIA (HI1)/FII(K) of *K. pneumoniae*. To the best of our knowledge, we present the first report on the isolation of the two isolates from tracheal lavage and urine-derived from different cats. Of concern, having found five CRKP, which also appeared in other countries was considered a severe situation [19,20]. In this study, carbapenem resistance genes, *bla*_OXA-181_ and *bla*_NDM-5_, were harbored in IncX3-type plasmids of different CRKP but no *bla*_OXA-48_ and *bla*_KPC_ was found in companion animals, while *bla*_OXA-48_ was prevalent in other countries [24]. Resistance genes, *bla*_CTX-M_ and *bla*_TEM_, of ESBL, showed a lower prevalence among companion animals, but the prevalence was higher than Japan (34.8%) [18], Italy (21.4%) [17], and other European countries (11.2%) [25]. In addition, CTX-M-genotypes were diverse in different species and countries. Herein, *bla*_CTX-M-65_ and *bla*_CTX-M-55_ predominated in the isolates of humans and companion animals, a phenomenon similar to findings from a previous study of pets in South Korea [19] and China [26]. Moreover, *K. pneumoniae* of type CTX-15 was prevalent in companion animals in Japan [18]. These findings demonstrate that the mutations of *K. pneumoniae* resistance genes are not limited to specific hosts or regions, and highlights the necessity of coordinated control in One Health. Furthermore, the genes *aac(6′)Ib-cr* and *oqxAB* of quinolone from companion animals showed higher resistance rates than those from human isolates; the rates were similar to those reported previously [18,27]. Otherwise, the most prevalent STs were ST1 and ST37, which concur with a previous report in China [26], but contrasts from reports in companion animals from Portugal [15] and Japan [18].

Based on the current understanding, virulence factors are encoded by several paragenes and can further increase the severity and/or pathogenicity of *K. pneumoniae* infection. Researches proved that any three of four siderophore systems (*ent*, *ybt*, *iuc*, and *iro*) could enhance virulence in murine models [3]. The *ybt* and *iuc* of human isolates were significantly more prevalent than that of companion animals. However, virulence factor-encoding genes of dogs and cats exhibited no difference in the current study. Previous reports had indicated that hvKP is mainly associated with ST23 and CRKP primarily belonging to ST11, which are considered the two major clinically important pathogens in China [28]. In this study, hvKP isolates (61.0%) from companion animals were less than the previously reported rates (76.4%) in humans [12]. Unfortunately, five ST23 isolates harbored all siderophore systems, identified as hvKP by *G. mellonella* model, but contained the least resistance genes. In companion animals, the four CR-hvKP were ST1 (*n* = 3) and ST16 (*n* = 1), whereas the CR-hvKP isolates were common genetic types such as ST11 and ST23 [14]. Hypermucoviscosity may be the most dominant virulence factor of *K. pneumoniae*, but its genetic basis and pathogenic factors are not direct. Due to the presence of one or two of the para-regulatory genes *rmpA* or *rmpA2*, this phenotype is usually associated with the overproduction of capsules [29]. Meanwhile, six of nine hypermucoviscous isolates were hvKP, but only one hypermucoviscous isolate in companion animals showed the MDR phenotype. Additionally, MDR-hvKP accounted for 68.3% of MDR isolates, 64.1% of hvKP, and 39.0% of total *K. pneumonia* isolates. The above findings demonstrate the high-risk of *K. pneumonia*, creating a huge treatment limitation in pet clinics.

In conclusion, the present study did a large-scale investigation of antimicrobial resistance and molecular genetic analysis in *K. pneumoniae* from clinical companion animals in Beijing, China. A high prevalence of MDR and hypervirulent *K. pneumoniae* isolates were found from dogs and cats. The wide distribution of amoxicillin-clavulanic and third-generation cephalosporins in veterinary hospitals may contribute to the ESBL resistance in these isolates. The presence of *mcr*, *bla*_OXA181_ and *bla*_NDM_ in *K. pneumoniae* demonstrates that the pathogen is a potential reservoir of colistin and carbapenem resistance genes in pet clinics. Meanwhile, the emergence of MDR-hvKP and epidemic clones elevates the risks of veterinarians; however, the predominant clones of CRKP are scarce in human-related ST clones. These findings emphasize the importance of managing *K. pneumonia* comorbidities and scientifically conducting antimicrobial susceptibility tests for more accurate treatments. This would reduce the spread of such high-risk clonal lineages to ensure the safety of companion animal practitioners and public health.

## 4. Materials and Methods

### 4.1. Samples Collection and Bacterial Characterization

All samples of companion animals were collected aseptically from the Veterinary Teaching Hospital of China Agricultural University (VTH-CAU), Lpet Veterinary Diagnostic Center (LVDC-Beijing), and North China (Tianjin) Testing Center, between July 2017 and October 2019. Sampling was conducted following the principles of the Beijing Municipality Review of Welfare and Ethics of Laboratory Animals and approved by the China Agricultural University Animal Ethics Committee document (No. AW01017102-2). *K. pneumoniae* isolates were isolated using 5% sheep blood agar and MacConkey Inositol Adonitol Agar medium (HopeBio, Qingdao, China) containing 100 mg/L carbenicillin following aerobic incubation at 37 °C overnight. The DNA of individual clones with the red centre was extracted by TIANamp Bacteria DNA Kit (Tiangen, Beijing, China) with the protocol as stipulated by the manufacturer. Subsequently, the DNA was used as templates for polymerase chain reaction (PCR) amplification of 16S rDNA gene sequencing (16SrRNA-F: AGAGTTTGATCCTGGCTCAG, 16SrRNA-R: ACGGCTACCTTGTTACGACTT) and conditions consisted 95 °C (10 min), 30 cycles of [95 °C (30 s), 55 °C (30 s), 72 °C (90 s)], and 72 °C (10 min) as previously described [30]. Amplicons were sequenced to reveal bacterial genus using the BLAST algorithm.

### 4.2. Antimicrobial Susceptibility Testing

For this experiment, we adopted agar/broth microdilution method with two-fold dilutions for the commonly used antimicrobial agents in human and/or companion animal clinics, including 16 antimicrobials belonging to 10 different categories, such as β-lactams combination agents (amoxicillin-clavulanate, piperacillin-tazobactam, and ceftazidime-avibactam), cephalosporins (cefotaxime and cefepime), carbapenems (meropenem and imipenem), monobactams (aztreonam), fluoroquinolones (ciprofloxacin and enrofloxacin), aminoglycosides (gentamicin and amikacin), tetracyclines (doxycycline), lipopeptides (colistin), phenicols (florfenicol), folate pathway antagonists (trimethoprim-sulfamethoxazole). All antibiotics were purchased from China Institute of Veterinary Drug Control and Solarbio (Beijing, China). Results of minimum inhibitory concentrations (MICs) were expressed according to breakpoint tables of Clinical and Laboratory Standards Institute (CLSI) documents VET08-ED4:2018/M100-ED30:2020 and European Commission on Antimicrobial Susceptibility Testing (EUCAST) documents (version 9.0, 2019). *E. coli* ATCC 25922 served as a quality control organism. Based on Standardized International Terminology; the drug resistance was classified as MDR for three or more categories of antimicrobial agents.

### 4.3. Mucoviscosity Assay

Because mucoviscous cells remain in suspension, whilst non-mucoid cells could form pellets after centrifugation, measurement of the turbidity after low-speed centrifugation can serve as an indicator of hypermucoviscosity. In short, bacterial isolates were grown in Lysogeny broth (LB) broth at 37 °C after 6 h incubation with shaking and then centrifuged at 1000× *g* for 5 min. The optical density at 600 nm (OD600) values of the supernatant were determined and measured. Mucoviscosity isolates were more difficult to pellet, so the supernatants have higher absorbance readings.

### 4.4. Galleria Mellonella Virulence Assay

Here, we tested the in vivo virulence using the *G. mellonella* infection model to indicate hypervirulence, as previously described [31]. In total, 10 randomly selected *G. mellonella* larvae approximately 250 to 350 mg for each isolate were purchased from Huiyude Biotech Company, Tianjin, China. Bacteria cells were cultured in a mid-log-phase and pelleted via centrifugation at 3500 rpm, washed twice, and resuspended in 0.01 M phosphate-buffered saline (PBS, pH6.5). Larvae were injected with 10 μL bacterial suspension (with 10^6^ colony-forming units, CFU) via the rear left proleg using a micro-sample syringe. PBS and ATCC43816 were used as negative and positive control groups, respectively. After injection, larvae were placed in 90 mm Petri-dishes and kept at 37 °C in the dark. Insects were considered dead when they did not respond to physical stimuli. We monitored death at 6 h intervals during 72 h. Experiments were performed in triplicate.

### 4.5. Whole-Genome Sequencing and Molecular Analysis

To extract the genomic DNA of the isolates, the TIANamp Bacteria DNA Kit was used following the manufacturer’s manuals. Indexed DNA libraries were constructed using the KAPA Hyper Prep Kit Illumina platforms (Roche, Basel, Switzerland) following the instruction, then sequenced on the Illumina Hiseq X Ten platform via the 150-bp paired-end strategy (Annoroad, Beijing, China). The draft genomes were assembled using SPAdes (version 3.9.0) [32]. All whole-genome sequencing data for this work are deposited in the GenBank and under BioProject accession no. PRJNA684769. Plasmid types, antibiotic resistance genes, and virulence genes were identified using abricate (https://github.com/tseemann/abricate, accessed on 13 August 2020), whereas the multilocus sequence typing (MLST) was obtained using the SRST2 toolkit (version 0.2.0) [33]. Capsule serotype (KL) and O-antigen (O) were analyzed using Kleborate (https://github.com/katholt/Kleborate, accessed on 16 September 2020). All draft genomes were used for core-genome alignments, after which we constructed a phylogenetic tree using parsnp in the Harvest package (version 1.1.2) [34]. The tree was visualized using the online tool Interactive Tree of Life (iTOL, http://itol.embl.de/, accessed on 19 September 2020) with the corresponding features of each isolate. The minimum spanning tree for all STs was generated by BioNumerics version 7.6 (Applied Maths, Sint-Martens-Latem, Belgium) using the BURST algorithm between different backgrounds. To compare the genetic context in the different plasmids, BLAST Ring Image Generator (BRIG) was applied [35]. Additionally, we analyzed the SRA of *K. pneumoniae* derived from humans in the NCBI database, which was collected in Beijing between July 2017 and October 2019.

### 4.6. Statistical Analysis

Statistical significance was determined using Chi-square (χ^2^) and Fisher’s exact test in SPSS Statistics (version 22, IBM Corporation, Armonk, NY, USA). The level of significance was set at *p* < 0.05.

## Figures and Tables

**Figure 1 pathogens-10-00271-f001:**
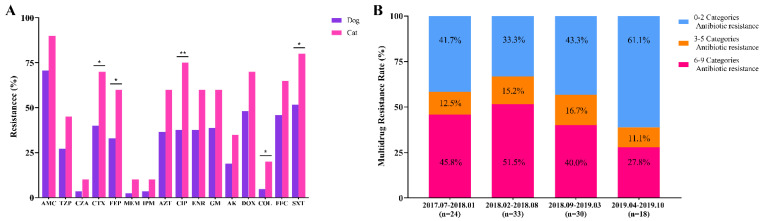
Antimicrobial resistance of *K. pneumoniae* isolates. (**A**) The resistance of isolates to different antibiotics from dogs and cats, respectively. AMC, Amoxicillin-clavulanate; TZP, Piperacillin-tazobactam; CZA, Ceftazidime-avibactam; CTX, Cefotaxime; FEP, Cefepime; MEM, Meropenem; IPM, Imipenem; AZT, Aztreonam; CIP, Ciprofloxacin; ENR, Enrofloxacin; GM, Gentamicin; AK, Amikacin; DOX, Doxycycline; COL, Colistin; FFC, Florfenicol; SXT, Trimethoprim-sulfamethoxazole. *p*-values as determined by Chi-square (χ2) and Fisher’s exact test in SPSS Statistics. * *p* < 0.05, ** *p* < 0.01. (**B**) Multi-drug-resistant rates of 105 isolates based on the antimicrobial categories from July 2017 to October 2019.

**Figure 2 pathogens-10-00271-f002:**
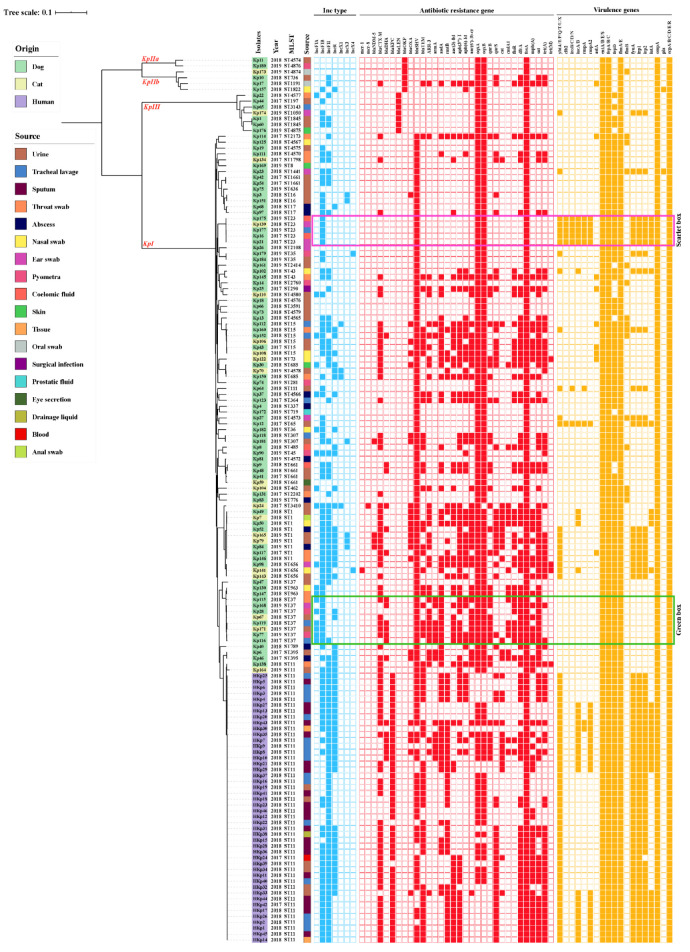
Distribution of *K. pneumoniae* phylogroups, year, multilocus sequence typing (MLST), Inc-type plasmid, antibiotic-resistance genes, and virulence-associated genes among isolates from companion animals and humans across the phylogenetic tree. Scarlet box: ST23-hvKP with the most virulence genes harboring the least resistance genes; green box: ST37-hvKP carrying only the common virulence factors while taking along relatively more resistance genes.

**Figure 3 pathogens-10-00271-f003:**
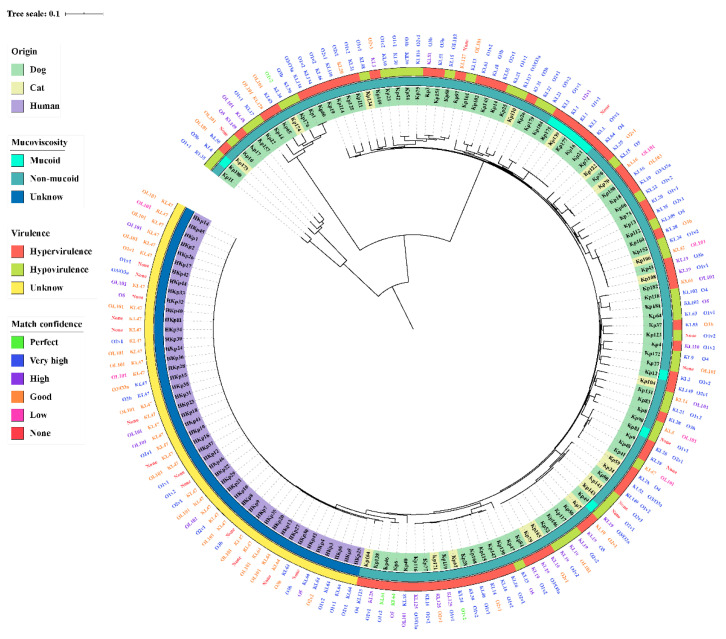
Distribution of *K. pneumoniae* mucoviscosity, virulence evaluation by *Galleria mellonella*, capsular serotype, and O-antigen type among isolates from companion animals and humans across the phylogenetic tree. The match confidences of serotypes were perfect, very high, high, good, low, and none.

**Figure 4 pathogens-10-00271-f004:**
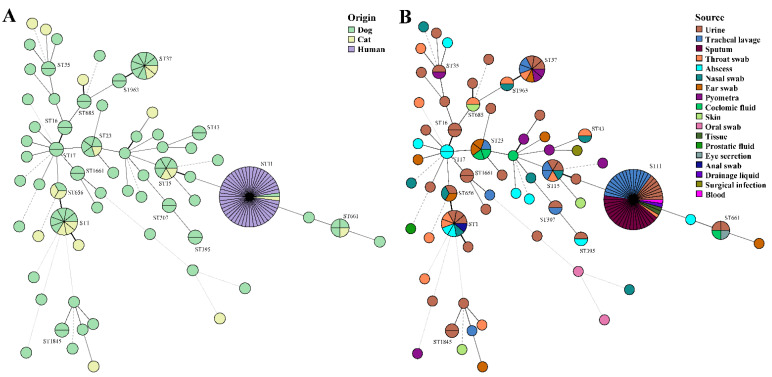
Minimum spanning trees of MLST typing. (**A**) STs colored based on different origins of dog, cat, and human. (**B**) STs colored based on different sources, such as urine, tracheal lavage, sputum, throat swab, abscess, nasal swab, ear swab, pyometra, coelomic fluid, skin, oral swab, tissue, prostatic fluid, eye secretion, anal swab, drainage liquid, surgical infection, and blood.

**Figure 5 pathogens-10-00271-f005:**
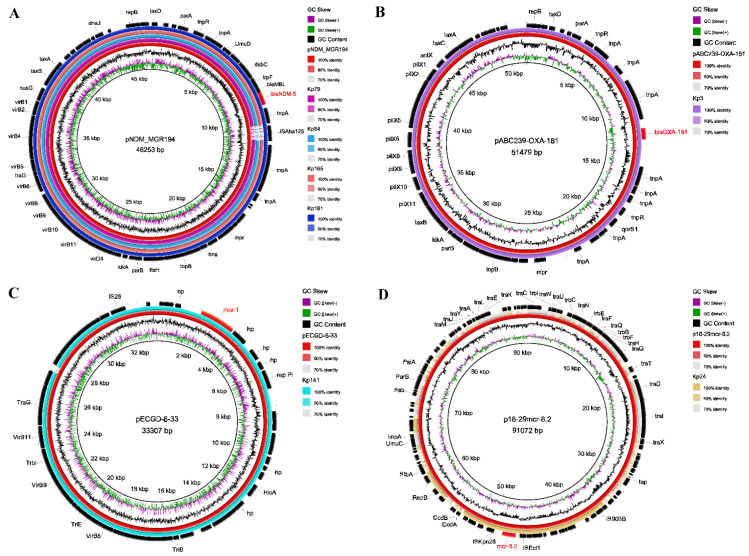
BLAST ring comparison of plasmids in *K. pneumoniae* isolates, each ring represents a separate isolate. The comparison of plasmids carrying (**A**) *bla*_NDM-5_ (IncX3), (**B**) *bla*_OXA-181_ (IncX3), (**C**) *mcr-1* (IncX4), and (**D**) *mcr-8* (IncFIA). The internal ring is the reference sequence of pNDM_MGR194 (46253bp, accession No. KF220657), pABC239-OXA-181 (51479bp), pECGD-8-33 (33307bp), and p18-29mcr-8.2 (91072bp) respectively, and the outside rings are isolates from this study, which are similar to the reference plasmid.

**Table 1 pathogens-10-00271-t001:** The distribution of *K. pneumoniae* samples and isolates.

Parameters	Category	No. of Samples (%)	Kp Isolates (%, 95% CI)
Origin	Dog	3356 (62.6)	85 (2.5, 2.0–3.1)
	Cat	2003 (37.4)	20 (1.0, 0.6–1.5)
Gender	Male	3352 (62.5)	58 (1.7, 1.3–2.2)
	Female	2007 (37.5)	47 (2.3, 1.7–3.1)
Source	Urine	2879 (53.7)	37 (1.3, 0.9–1.8)
	Throat swabs	200 (3.7)	13 (6.5, 3.5–10.9)
	Nasal swabs	194 (3.6)	9 (4.6, 2.1–8.6)
	Abscess	391 (7.3)	10 (2.6, 1.2–4.7)
	Tracheal lavage	126 (2.4)	8 (6.3, 2.8–12.1)
	Ear swabs	267 (5.0)	8 (3.0, 1.3–5.8)
	Pyometra	158 (2.9)	7 (4.4, 1.8–8.9)
	Coelomic fluid	367 (6.8)	4 (1.1, 0.3–2.8)
	Skin	452 (8.4)	3 (0.7, 0.1–1.9)
	Oral swabs	21 (0.4)	2 (9.5, 1.2–30.4)
	Prostatic fluid	22 (0.4)	1 (4.5, 0.1–22.8)
	Surgical infection	70 (1.3)	1 (1.4, 0–7.7)
	Anal swabs	32 (0.6)	1 (3.1, 0.1–16.2)
	Eye secretion	41 (0.8)	1 (2.4, 0.1–12.9)
	Others	139 (2.6)	0 (0, 0–2.6)
Total	——	5359	105 (2.0, 1.6–2.4)

## Data Availability

All whole-genome sequencing data for this work are deposited in the GenBank and under BioProject accession no. PRJNA684769.

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
