# Peer review of "Molecular Investigation of Klebsiella pneumoniae from Clinical Companion Animals in Beijing, China, 2017–2019"

_pathogens, 2021, doi:10.3390/pathogens10030271_

Round 1

Reviewer 1 Report

The manuscript "Molecular Investigation of Klebsiella pneumoniae from clinical companion animals in Beijing, China, 2017-2019" details the isolation of pathogenic Klebsiella pneumoniae from close human pets (cats and dogs) and molecular study of virulence factors. The manuscript adequately combines field and lab work with molecular investigation techniques. There are just few minor corrections to be made:

  1. Kindly check the language. For example, the first few words in the abstract "This work aimed to elucidate the ......", should be "This work is aimed to elucidate the ......". There are several such instances. So keenly re-read the manuscript to fix similar errors.
  2. Due to the plethora of data, Figure 2 and 3 are too much compressed to read. Kindly provide a high resolution downloadable figure in the final form of the manuscript so that the readers are able to understand the data presented in these figures.
  3. Briefly describe the 16S primers and PCR conditions so that the readers do not have to download another article to find those out.
  4. As the details of the Figure 5 are explained in the captions, please provide brief details in captions for the rest of the figures.

Author Response

Response to Reviewer 1 Comments

The manuscript "Molecular Investigation of Klebsiella pneumoniae from clinical companion animals in Beijing, China, 2017-2019" details the isolation of pathogenic Klebsiella pneumoniae from close human pets (cats and dogs) and molecular study of virulence factors. The manuscript adequately combines field and lab work with molecular investigation techniques. There are just few minor corrections to be made:

Point 1: Kindly check the language. For example, the first few words in the abstract "This work aimed to elucidate the ......", should be "This work is aimed to elucidate the ......". There are several such instances. So keenly re-read the manuscript to fix similar errors.

Response 1: Many thanks for your careful review. We apologies for our mistakes and modified these errors in the revised manuscript. 

Point 2: Due to the plethora of data, Figure 2 and 3 are too much compressed to read. Kindly provide a high resolution downloadable figure in the final form of the manuscript so that the readers are able to understand the data presented in these figures.

Response 2: We have provided the highest resolution figures in the revised manuscript and upload them into submitting system according to your suggestion.

Point 3: Briefly describe the 16S primers and PCR conditions so that the readers do not have to download another article to find those out.

Response 3: We have added the primers sequencing of 16SrRNA and the PCR conditions in the revised manuscript (Lines 348-351).

Point 4: As the details of the Figure 5 are explained in the captions, please provide brief details in captions for the rest of the figures.

Response 4: We have added brief details in the captions of the Figure 5 in the revised manuscript (Lines 231-235).

Reviewer 2 Report

The manuscript “Molecular investigation of Klebsiella pneumoniae from clinical companion animals in Beijing, China, 2017-2019" addresses a very important issue on the prevalence of antibiotic resistance and virulence genes among K. pneumoniae isolated from companion pets in China. Moreover, the authors determined the genotypes of all tested isolates and characterised their plasmids.

In my opinion, the premise of this manuscript is very interesting. This paper is well written and has a good body of different analyses, but some of the results described  are not explained enough. First of all, throughout the manuscript, there is little attempt to disentangle the association between hypermucoviscous and hypervirulent isolates of Klebsiella pneumonia. Which variants, mucoid or non-mucoid are more resistant to antibiotics, or are more virulent. What is the difference between hypervirulence and hypovirulence presented on figure 3? Second, what is association between capsule serotype or O-antigen type and a hypermucoviscous phenotype?

Furthermore, I think that the mucoviscousity in Klebsiella should not be tested by the string test. This test has many caveats, it is not quantitative and shows repeatability issues because it is strongly dependent on a number of conditions, including temperature, media, etc. A more objective way of assessing mucoviscosity in Klebsiella is the sedimentation assay, in which after slow centrifugation, mucoviscous cells remain in suspension, whilst non-mucoid cells are able to pellet. Please re-test your clones.

Minor comments:

Figure 1b requires more explanation. Particularly what does the "categories antibiotic resistance" means or how they were calculated.

L118: expand the SRA shortcut, please

L121: I presume that in the genomes of bacteria isolated from human, not in human genomes. Check it throughout the manuscript, please.

L292: The sentence about the positive control of the string test is too obvious. Please, delete it.

L329-332: Provide the sources for antibiotics (company, country), please.

All legends and descriptions on the figures have too small fonts. They are poorly visible.

Author Response

Response to Reviewer 2 Comments

The manuscript “Molecular investigation of Klebsiella pneumoniae from clinical companion animals in Beijing, China, 2017-2019" addresses a very important issue on the prevalence of antibiotic resistance and virulence genes among K. pneumoniae isolated from companion pets in China. Moreover, the authors determined the genotypes of all tested isolates and characterised their plasmids.

In my opinion, the premise of this manuscript is very interesting. This paper is well written and has a good body of different analyses, but some of the results described are not explained enough.

Point 1: First of all, throughout the manuscript, there is little attempt to disentangle the association between hypermucoviscous and hypervirulent isolates of Klebsiella pneumonia. Which variants, mucoid or non-mucoid are more resistant to antibiotics, or are more virulent.

Response 1: Thanks a lot for your positive comments. We further conducted a more in-depth analysis of virulence results according to your suggestion (Lines 158-159). In brief, there was no association between hypervirulent K. pneumonia (hvKP) and hypermucoviscous K. pneumonia (hmKP) (p > 0.05). The resistance and virulence of mucoid and non-mucoid isolates had no significant difference in this study (p > 0.05).

Point 2: What is the difference between hypervirulence and hypovirulence presented on figure 3?

Response 2: We tested the in vivo virulence using the Galleria mellonella infection model as an indication of hypervirulence. Sixty-four strains demonstrated higher or equal pathogenicity to the positive control ATCC43816 after injection for 72 h in G. mellonella, demonstrating that these isolates were hvKP. The remaining 41 strains were defined as hypovirulence. Virulence of humans K. pneumoniae obtained from NCBI were unknow in Figure 3. There was no significant difference in antibiotic resistance between hypervirulence and hypovirulence strains from companion animals (p > 0.05) (Lines 160-161).

Point 3: Second, what is association between capsule serotype or O-antigen type and a hypermucoviscous phenotype?

Response 3: Many thanks for your constructive comments. The capsule serotype and O-antigen type had some certain relationship with hypermucoviscous. In this study, the KL1 and KL2 isolates were hmKP, and the O-antigen type of 77.8% (n=7/9) hmKP was O1v2 (Lines 166-167).

Point 4: Furthermore, I think that the mucoviscousity in Klebsiella should not be tested by the string test. This test has many caveats, it is not quantitative and shows repeatability issues because it is strongly dependent on a number of conditions, including temperature, media, etc. A more objective way of assessing mucoviscosity in Klebsiella is the sedimentation assay, in which after slow centrifugation, mucoviscous cells remain in suspension, whilst non-mucoid cells are able to pellet. Please re-test your clones.

Response 4: Thanks for your valuable suggestions. We have performed low-speed centrifugation to further differentiate the subtle difference in the levels of capsule production in different strains. We agree with you that the bacteria with thick and mucoid capsules were more difficult to pellet. We have revised the corresponding results (Lines 156-157) and the specific experimental operation (Line 374-381).

Minor comments:

Point 1: Figure 1b requires more explanation. Particularly what does the "categories antibiotic resistance" means or how they were calculated.

Response 1: We have added the details to explain this expression in Lines 360-365.

Point 2: L118: expand the SRA shortcut, please.

Response 2: We added the full name of “Sequence Read Archive (SRA)” in Line 116.

Point 3: L121: I presume that in the genomes of bacteria isolated from human, not in human genomes. Check it throughout the manuscript, please.

Response 3: We have modified the error description in the revised manuscript (Line 120).

Point 4: L292: The sentence about the positive control of the string test is too obvious. Please, delete it.

Response 4: We have deleted it in the revised manuscript.

Point 5: L329-332: Provide the sources for antibiotics (company, country), please.

Response 5: All antibiotics were purchased from China Institute of Veterinary Drug Control and Solarbio (Beijing, China). The description has been added in Lines 366-367 of the revised manuscript.

All legends and descriptions on the figures have too small fonts. They are poorly visible.

Response: We are sorry for our negligence. We have enlarged the adjustable partial font size of the legends and uploaded the highest resolution downloadable figures for review in the revised manuscript and submitting system according to your suggestion.

Round 2

Reviewer 2 Report

I have read a point-by-point response to my comments. I found this manuscript improved, although I have still one comment. I did found no data supporting mucoviscousity. Please make this data available. 
